# BATF alleviates ox-LDL-induced HCAEC injury by regulating SIRT1 expression in coronary heart disease

Bei Tian[1], Jingyu Ji[1], Can Jin📧[2]*

1 Nursing Department, Shanghai University of Medicine and Health Sciences Affiliated Zhoupu Hospital, Shanghai, China, 2 Department of Cardiovascular Medicine, Shanghai University of Medicine and Health Sciences Affiliated Zhoupu Hospital, Shanghai, China

* jincan0722@126.com

## Abstract

**Data Availability Statement:** All relevant data are within the manuscript and its Supporting information files.

**Funding:** Minsheng Research Project of Pudong New Area Science and Technology Development

### Background

Coronary heart disease (CHD) represents a significant global health concern, arising from an intricate interplay between genetic predisposition and environmental influences, with a pivotal involvement of oxidized low-density lipoprotein (ox-LDL) in the pathophysiology of it. We aimed to elucidate the synergistic dynamics of B cell activating transcription factor (BATF) and Sirtuin 1 (SIRT1) in cell injury caused by ox-LDL, reveal potential therapeutic strategies for CHD.

### Methods

The GSE42148 dataset was used to analyze Differentially expressed genes (DEGs) to construct a gene co-expression network. Then bioinformatics analysis was performed on key modules to select the BATF gene. *In vitro* experiments were conducted to investigate the protective impact of BATF against human coronary artery endothelial cells (HCAEC) injury induced by ox-LDL. Further investigations probed the synergistic impact of BATF and SIRT1 modulation on cellular apoptosis and damage in the presence of ox-LDL.

### Results

BATF was significantly down-regulated in the CHD sample of the GSE42148 dataset. *In vitro* assays have proven that BATF alleviates ox-LDL-induced HCAEC injury. Notably, BATF emerged as a pivotal regulator of SIRT1 expression post ox-LDL exposure. Subsequent experiments underscored the interplay between BATF and SIRT1 in mitigating ox-LDL-induced apoptosis and Lactate Dehydrogenase (LDH) activity elevation, highlighting their collaborative role in cellular protection.

Fund (PKJ2023-Y13) Key Discipline Group of Pudong New Area Health Commission (PWZxq2022-11) Peak Discipline Construction of Pudong New Area Health Commission (PWYgf2021-04) Key Sub-specialty of Pudong New Area Health Commission (PWZy2020-08) We received key support from funders above for this study. Funding financial support from the Pudong Science and Technology Development Fund and the Pudong Health and Wellness Committee provided the necessary financial security to enable the study to proceed smoothly. These funds were used for study design, data collection and analysis, and preparation of relevant manuscripts. The support of the funders was an important guarantee that this study could make progress and achieve results. We sincerely thank them for their generous support and trust.

**Competing interests:** The authors have declared that no competing interests exist.

## Conclusion

The research findings suggested a prospective protective function of BATF in HCAEC injury induced by ox-LDL, likely through the mediation of SIRT1 regulation. These results could offer fresh perspectives on the etiology of CHD and possible treatment avenues.

## Introduction

Coronary heart disease (CHD) arises from the interplay of genetic and environmental factors, contributing to its multifactorial nature [1,2]. Characterized by symptoms such as angina, dyspnea, chest tightness, dizziness, fatigue, and cough, quality of life is significantly impacted by CHD in patients [3,4]. Lifestyle choices, including smoking, sedentary behavior, unhealthy dietary habits, and excessive alcohol consumption, significantly contribute to disease etiology [5–7]. Biomedical factors, including hypertension, dyslipidemia, diabetes, and obesity, play a crucial role in disease development [8,9]. Advanced age, male gender, and familial history of CHD additionally amplify the risk of the condition [10,11]. Globally, the prevalence of CHD is alarmingly high, with epidemiological data indicating an ascending trend in both incidence and mortality rates [12]. Clinically, CHD management is multifaceted, with pharmacological interventions such as antiplatelet agents, statins, beta-blockers, ACE inhibitors, and nitrates playing a pivotal role [13–15]. Each drug class possesses a unique mechanism that targets different aspects of the disease pathology. By elucidating the complex pathophysiology of CHD, we can create conditions that will allow creative preventative techniques to be developed and more efficacious treatment paradigms, thereby improving the overall prognosis for CHD patients.

Cardiovascular disease, especially CHD, is a major health problem worldwide [16]. Oxidized low-density lipoprotein (ox-LDL) has emerged as a critical element in the development of atherosclerosis, a major underlying factor in CHD [10,17]. In addition, ox-LDL had been shown to induce damage in human coronary artery endothelial cells (HCAEC), leading to endothelial dysfunction, a key early event during the progression of atherosclerosis [18–20]. Furthermore, It has been proposed that ox-LDL may initiate apoptosis and accelerate the development of CHD [19,21]. In the cellular environment, multiple factors work together to maintain cellular homeostasis and resist ox-LDL-induced damage [22]. For example, BATF had emerged as a potential protective factor against ox-LDL-induced damage [23]. BATF was known to regulate various immune responses, but its role in CHD is unknown. Recent investigations have proposed that BATF, a transcription factor, may engage in interactive dynamics with SIRT1, a NAD$^+$-dependent protein deacetylase, which was pivotally associated with cardiovascular health [24]. SIRT1 is known to exert a protective role against inflammation and oxidative stress, which are important elements in atherosclerosis development [25]. Therefore, gaining insight into the interaction between SIRT1 and BATF in the setting of ox-LDL-induced HCAEC damage may help to clarify the pathophysiology and provide novel treatment approaches for CHD.

In light of the escalating global burden of CHD and its complex etiological nexus, there is an imperative need to delineate the underlying molecular pathways contributing to its progression. Building on the established role of ox-LDL in atherosclerosis and its potential to induce damage in HCAEC, our study aims to unravel the intricate relationship between BATF, a transcription factor, and SIRT1, a protective NAD+-dependent protein deacetylase. While both molecules have been individually associated with cellular defense mechanisms against

oxidative stress, their synergistic dynamics, especially in the realm of ox-LDL-induced cellular injury, remain largely uncharted. Our endeavor seeks to elucidate this interplay, offering potential insights into novel therapeutic avenues for CHD, ultimately enhancing patient outcomes.

## Material and methods

### Download and analysis of the GSE42148 gene expression profile

The CHD-associated gene expression profile GSE42148 was obtained from the Gene Expression Omnibus (GEO) database (https://www.ncbi.nlm.nih.gov/geo/) in the series matrix file format. This dataset included gene expression profiles from 11 normal samples, which served as the control group, and 13 CHD samples, which were treated as the case group. To identify the differentially expressed genes (DEGs) of the two groups, we utilized GEO2R, an interactive online tool designed for comparing sample groups within a GEO series. In this case, the gene expression profiles of the case group and the control group were compared using it. DEGs were selected based on a fold change (FC) > 1.3 (up-regulated DEGs) or < 0.7 (down-regulated DEGs), with a $p$-value < 0.05, to ensure the statistical importance of the variations in gene expression. Upon identifying the DEGs, we proceeded with data visualization using the "ggplot2" package in R software.

### Construction and analysis of gene co-expression network

To investigate the patterns of gene correlation in microarray data, the "Weighted gene co-expression network analysis (WGCNA)" package of R software screened 2044 DEGs from the GSE42148 gene expression profile and created a gene co-expression network for these DEGs. To build the scale-free network, we initially identified the optimal soft threshold power (β), which emphasizes strong correlations over weak ones and ensures scale-free topology. The selection of soft-threshold power was guided by the criteria for achieving an approximate scale-free topology, with a scale-free $R^2$ cutoff set at 0.85. After this, the automated network-building feature was used to carry out module identification and one-step network creation. Cluster dendrograms were then generated to visualize co-expressed modules, facilitating comprehensive analysis across all identified modules. Module eigengenes (ME), representing the first principal components of a given module, were computed to summarize gene expression profiles within each module. A feature gene adjacency heatmap was subsequently generated to visualize the relationship between modules. Following the calculation of the correlation between ME and the samples in case and control groups within the GSE42148 dataset, the two modules exhibiting the highest correlation were selected for subsequent analysis.

### Enrichment analysis of key modules and protein-protein interaction (PPI) network construction

Genes in the black and turquoise modules found by WGCNA of the GSE42148 gene expression profile were subjected to Wikipathway enrichment analysis using the "Cluster Profiler" program in R software. Pathways exhibiting $p$-values < 0.05 were selected, indicating a considerable enrichment. Subsequently, PPI network analysis was performed on the genes in the two modules utilizing the search tool for the retrieval of interacting genes/proteins (STRING, https://string-db.org/) database and Cytoscape software. To identify key genes in PPI networks, we sequentially used the BottleNeck, Degree, and EcCentrity algorithms in Cytoscape software to select the top 30 genes for visualization.

## Expression validation of overlapping genes and assessment of diagnostic accuracy

The "VennDiagram" package of R software was used to detect overlapping genes in the Bottle-Neck, Degree, and EcCentricity networks. Subsequently, the expression of these overlapping genes in two sample groups (cases and controls) in the GSE42148 gene expression profile was validated. For additional examination, genes with statistically significant variations in expression ($p < 0.05$) were taken into account. Subsequently, receiver operating characteristic (ROC) curve analysis was carried out for each gene using the "timeROC" program in R software to assess the possible clinical diagnostic significance of these overlapping genes in CHD. Also, the calculation of the area under the ROC curve (AUC) was performed, providing a quantitative measure of genetic diagnostic accuracy. The higher AUC value indicated the enhanced diagnostic ability of the gene for CHD.

## Cell culture and transfection

HCAECs were obtained from the Wuhan culture collection and cultured in endothelial cell medium supplemented with 10% FBS, 100 U/mL penicillin, and 100 μg/mL streptomycin at 37˚C in a humidified atmosphere with 5% $CO_2$. Cell confluence of 70–80% was achieved before the initiation of any treatment. To simulate the pathological conditions observed in CHD, 100 μg/mL of ox-LDL was used to treat the cells at a specific concentration for a predetermined duration. For gene manipulation, transfections were performed to either overexpress or knock down the BATF gene in the HCAECs. BATF overexpression was achieved by using specific plasmids carrying the BATF gene, while BATF knockdown was carried out with the application of two distinct short interfering RNAs (si-BATF#1 and si-BATF#2) designed to specifically target BATF mRNA, both following the manufacturer's directions for utilizing the Lipofectamine 2000 transfection reagent. HCAEC were treated with SIRT1 inhibitor (EX527) 100 μM and activator (S1129) 100 μM, respectively. A control group was treated with dimethyl sulfoxide (DMSO) at a final concentration of no more than 0.5%.

## Quantitative reverse transcription polymerase chain reaction (qRT-PCR) assay

After transfection and treatment, total RNA was extracted from the HCAECs using TRIzol reagent by the manufacturer's instructions. A NanoDrop spectrophotometer was used to evaluate the purity and amount of the extracted RNA, guaranteeing an A260/A280 ratio of 1.8 to 2.0. A High-Capacity cDNA Reverse Transcription Kit was then used to create complementary DNA (cDNA) from the total RNA. The mRNA levels of SIRT1 and BATF in the cells were assessed by qRT-PCR utilizing a Real-Time PCR System and SYBR Green PCR Master Mix. Primers for BATF, SIRT1, and the GAPDH were designed and synthesized. Primer sequences were displayed in Table 1. Ultimately, the $2^{-\Delta\Delta Ct}$ techniques were utilized to compute the relative expression.

## Western Blotting (WB) assay

Protein was extracted from the ox-LDL-treated HCAECs using phosphatase and protease inhibitors added to RIPA buffer. The BCA protein assay kit was used to measure the quantities of proteins. Equal protein amounts were separated on 10% SDS-PAGE and transferred onto a PVDF membrane. Subsequently, the membrane was blocked with 5% non-fat milk in TBST (Tris-buffered saline, 0.1% Tween 20) for 1 hour at room temperature and then incubated overnight at 4˚C with specific primary antibodies against BATF (Abcam, 1:1000), SIRT1

**Table 1. Primer sequences for qRT-PCR assay.**

| Gene Name | Sequence |
|---|---|
| BATF | F: 5′-GGCGCTAGCGCCACCATGCCTCACAGCTCCGAC-3′ |
| | R: 5′-GCCCTCGAGTCAGGGCTGGAAGCGC-3′ |
| SIRT1 | F: 5′-GCAGATTAGTAGGCGGCTTG-3′ |
| | R: 5′-TCTGGCATGTCCCACTATCA-3′ |
| GAPDH | F: 5′-CCAAGGAGTAAGAAACCCTGGAC-3′ |
| | R: 5′-GGGCC GAGTTGGGATAGGG-3′ |

Note: F represents forward primer and R represents reverse primer.

(Abcam, 1:1000), Bcl-2 (Abcam, 1:1000), Bax (Abcam, 1:1000), and GAPDH (Abcam, 1:1000). After primary antibody incubation, the membrane was washed and incubated with HRP-conjugated secondary antibodies for 1 h at room temperature. Protein bands were visualized using an enhanced chemiluminescence detection system, with GAPDH serving as the normalization control. All original uncropped Western blot images are available in S1 Raw images.

## Flow cytometry assay

Following ox-LDL treatment, BATF overexpression or knockdown, and the addition of SIRT1 activators or inhibitors, the Annexin V-FITC/PI Apoptosis Detection Kit was used to analyze apoptosis in HCAECs by the manufacturer's instructions. In brief, the cells were taken out and resuspended in a binding buffer after being cleaned with cold phosphate-buffered saline (PBS). For 15 minutes at room temperature, the cell suspension was incubated in the dark with Annexin V-FITC and propidium iodide (PI). A flow cytometer was used to examine apoptotic cells right away.

## Lactate Dehydrogenase (LDH) assay

LDH activity, as an indicator of cytotoxicity and membrane integrity, was measured in supernatants of ox-LDL-treated HCAECs overexpressed or knocked down with BATF and added SIRT1 activators/inhibitors. As directed by the manufacturer, the assay was carried out using an LDH Cytotoxicity Assay Kit. Following the manufacturer's instructions, LDH activity was computed and absorbance was measured using a microplate reader at 490 nm.

## In silico prediction of BATF binding sites

The JASPAR database (http://jaspar.genereg.net/) was used to estimate the probable binding locations of BATF in the SIRT1 promoter region, which is an open-access resource for predicting transcription factor binding sites. The core sequence and matrix similarity thresholds were set as recommended by the tool.

## Dual-luciferase reporter assay

To investigate the interplay between BATF and the 3' UTR of SIRT1, dual-luciferase reporter assays were conducted. Cells were co-transfected with either wild-type (WT) SIRT1 3'UTR or mutant constructs, alongside BATF overexpression or knockdown plasmids. Subsequently, luciferase activity was measured using the Dual-Luciferase Reporter Assay System following the manufacturer's protocol, with firefly luciferase activity normalized to Renilla luciferase activity for each sample.

## Chromatin immunoprecipitation (ChIP) assay

A ChIP assay was conducted to assess the binding affinity of BATF to the SIRT1 promoter using a ChIP Assay Kit as per the manufacturer's instructions. Briefly, cross-linked chromatin was sheared and subjected to immunoprecipitation with a BATF antibody. The purified DNA was then amplified via PCR using primers flanking the predicted BATF binding sites within the SIRT1 promoter.

## Statistical analysis

The information was displayed as the average ± standard deviation (SD) of a minimum of three separate trials carried out in duplicate. Student's t-test was used to examine the difference between the two groups, and one-way analysis of variance (ANOVA) and Tukey's post hoc test were used to compare data across many groups. SPSS software version 26.0 (IBM, Armonk, NY, USA) and GraphPad Prism 8.0 (GraphPad Software, La Jolla, CA, USA) were used for all statistical analyses. Statistical significance was defined as a $p$-value of less than 0.05.

## Results

### Identification of key modules associated with CHD in GSE42148 gene expression profile

From the 24 samples of the GSE42148 gene expression profile, 2044 DEGs in total were found, of which 1057 were up-regulated and 987 down-regulated (Fig 1A). To create a gene co-expression network, WGCNA was used to the discovered DEGs. The research yielded a value of 28 for the soft-thresholding power β, which was found to be the most ideal value to guarantee a network free of scale (Fig 1B). The hierarchical clustering of the DEGs, as depicted in the Cluster Dendrogram, showed a total of 8 color-coded modules (excluding the grey module), each representing a cluster of highly interconnected genes (Fig 1C). The modules were further analyzed through an eigengene adjacency heatmap, providing a visual representation of the relationships among the modules (Fig 1D). The correlation between gene modules and the two sample groups in the GSE42148 dataset was assessed (Fig 1E). The findings indicated that the black module exhibited the strongest association with the samples, followed by the turquoise module. The genes within these modules might be critical in the pathogenesis of CHD, providing potential targets for further investigation.

### Wikipathway enrichment analysis and PPI network of genes in black and turquoise modules

In a systematic exploration of gene pathway enrichments, the "Cluster Profiler" was employed to investigate the Wikipathway enrichments for genes identified within the black and turquoise modules, respectively. As delineated in Fig 2A, genes within the black module (n = 75) were predominantly enriched in several pathways, including Oligodendrocyte Specification and Differentiation, yielding Myelin Components for the Central Nervous System (CNS) (WP4304), Molecular Mechanisms of SARS-CoV-2 and Angiotensin-Converting Enzyme 2 Receptor (WP4883), Glial Cell Differentiation (WP2276), and MECP2 and Associated Rett Syndrome (WP3584), among others. Conversely, as depicted in Fig 2B, genes within the turquoise module (n = 371) primarily exhibited enrichment in pathways such as Nanomaterial-Induced Apoptosis (WP2507), Integrated Breast Cancer Pathway (WP1984), Apoptosis Modulation by HSP70 (WP384), and Target of Rapamycin (TOR) Signaling (WP1471), along with additional pathways. These results provided a direction for further exploring the functions of genes in these modules. Subsequently, the PPI networks for the genes within these modules

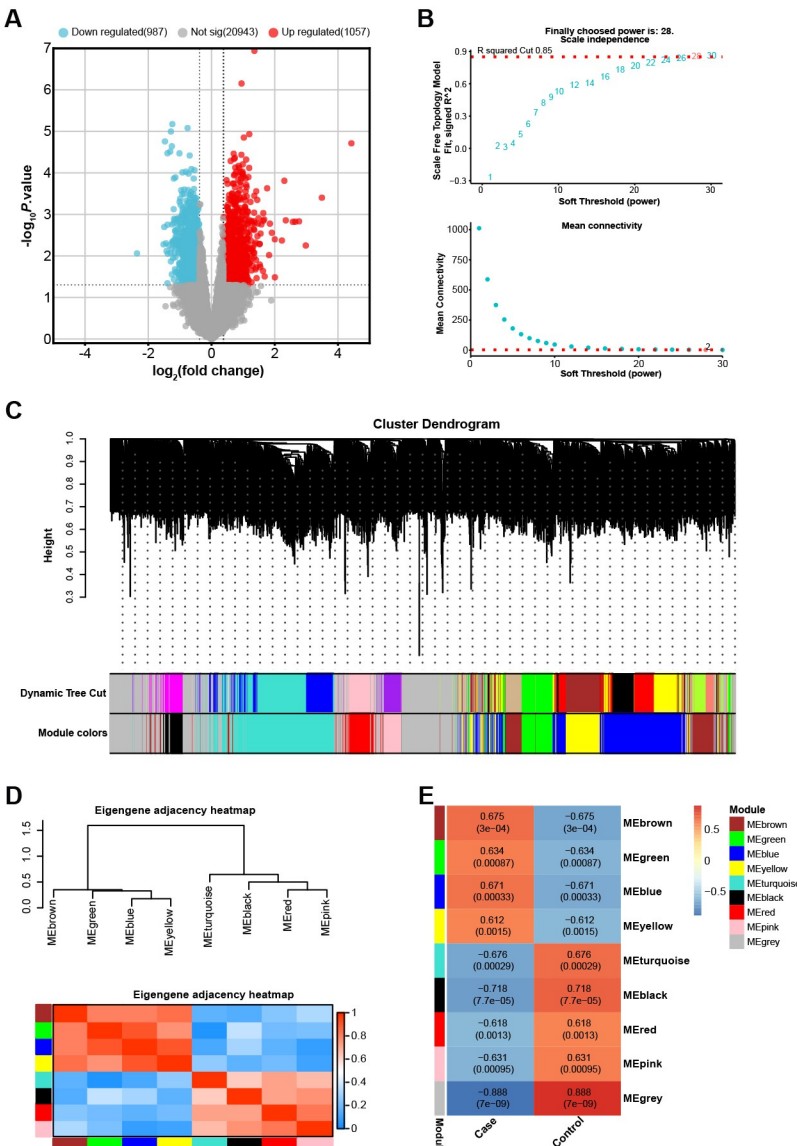

**Fig 1. Screening of DEGs and analysis of co-expression network in GSE42148 gene expression profile.** (A) Volcano plot of the 2044 DEGs identified from the GSE42148 gene expression profile, showing 1057 up-regulated and 987 down-regulated DEGs. (B) Analysis of the scale-free fit index for various soft-thresholding powers (β), with β = 28 selected as the most optimal value ensuring a scale-free network. (C) Cluster dendrogram of the DEGs, depicting a total of 8 color-coded modules (excluding the grey module), each representing a cluster of highly interconnected genes. (D) Eigengene adjacency heatmap illustrating the relationships among the identified gene modules. (E) Heatmap of the correlation between the gene modules and the two sample groups in the GSE42148 dataset, orange indicates a positive correlation and blue indicates a negative correlation.

were constructed and analyzed using Cytoscape software. Three different algorithms (Bottle-Neck, Degree, EcCentricity) in Cytoscape were used to identify the key nodes within these networks. The top 30 genes within these networks were visualized, revealing their interconnections and highlighting their potential significance in the pathogenesis of CHD (Fig 2C–2E). The functions of these discovered genes need to be confirmed by more validation and experimental research.

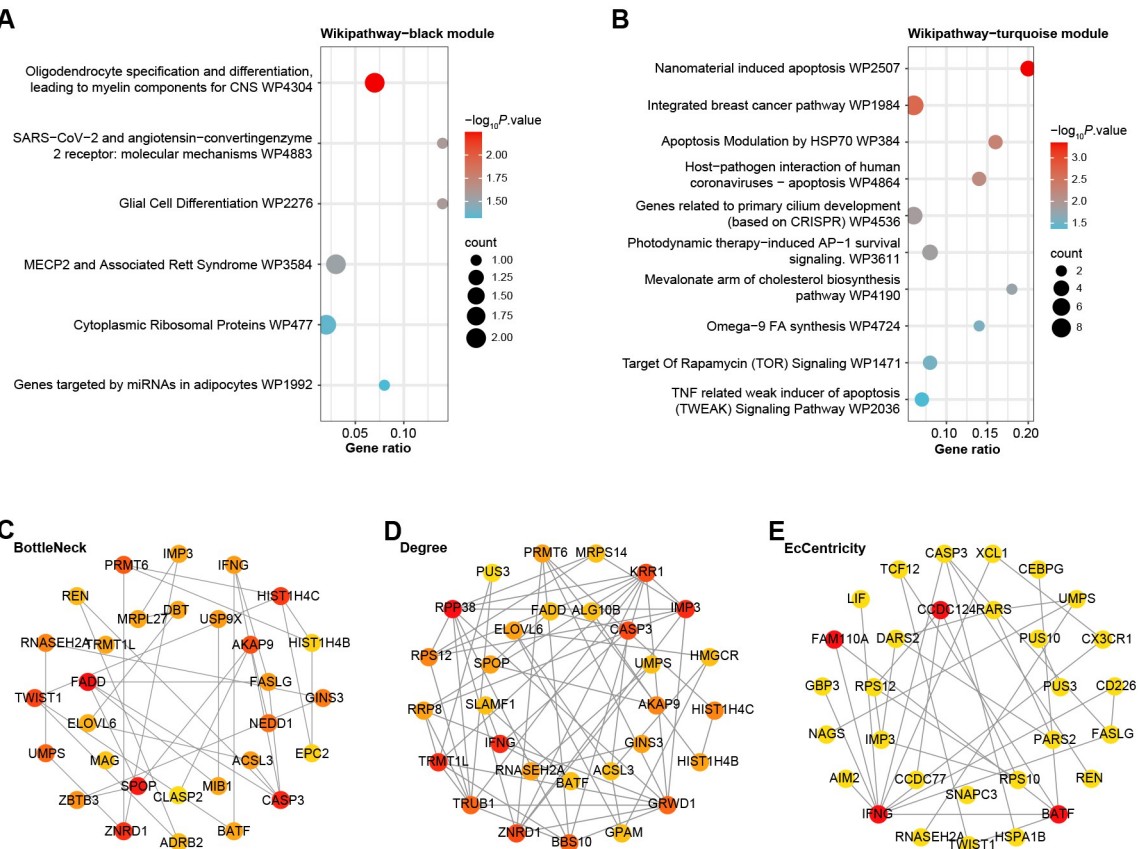

**Fig 2. Enrichment analysis and PPI network analysis for key modules.** (A-B) Bubble plots, Wikipathway enrichment analysis of genes in (A) the black module and (B) the turquoise module, the size of each bubble represents the gene count and the color represents the adjusted *p*-value. (C-E) Visualization of the PPI network of the top 30 genes identified by the BottleNeck (C), Degree (D), and EcCentricity (E) algorithms. In each network, nodes represent genes, edges represent protein-protein interactions, and the node size reflects the connection degree within the network.

## Identification of overlapping genes with potential clinical diagnostic value for CHD

After analyzing the genes in the PPI networks generated by the three algorithms, six overlapping genes were identified, namely IMP3, IFNG, CASP3, RNASEH2A, UMPS, and BATF (Fig 3A). The levels of expression of these six genes were then assessed using two GSE42148 sample groups. Each of the six genes was considerably downregulated in the case group, as Fig 3B demonstrated, indicating a possible link between the expression of these genes and CHD. All six genes had AUC values larger than 0.7, according to the ROC curve study results. BATF had a noteworthy AUC value of 0.89 (Fig 3C–3H), suggesting both its potential as a molecular marker and its strong clinical diagnostic significance.

## BATF alleviates ox-LDL-induced damage in HCAEC cells

BATF mRNA expression in HCAEC was observed to be reduced after ox-LDL treatment, according to qRT-PCR results (Fig 4A). This downregulation was further confirmed at the protein level by the WB assay (Fig 4B). Following the ox-LDL treatment, qRT-PCR was used to assess the effectiveness of BATF knockdown and overexpression in HCAECs (notably, the

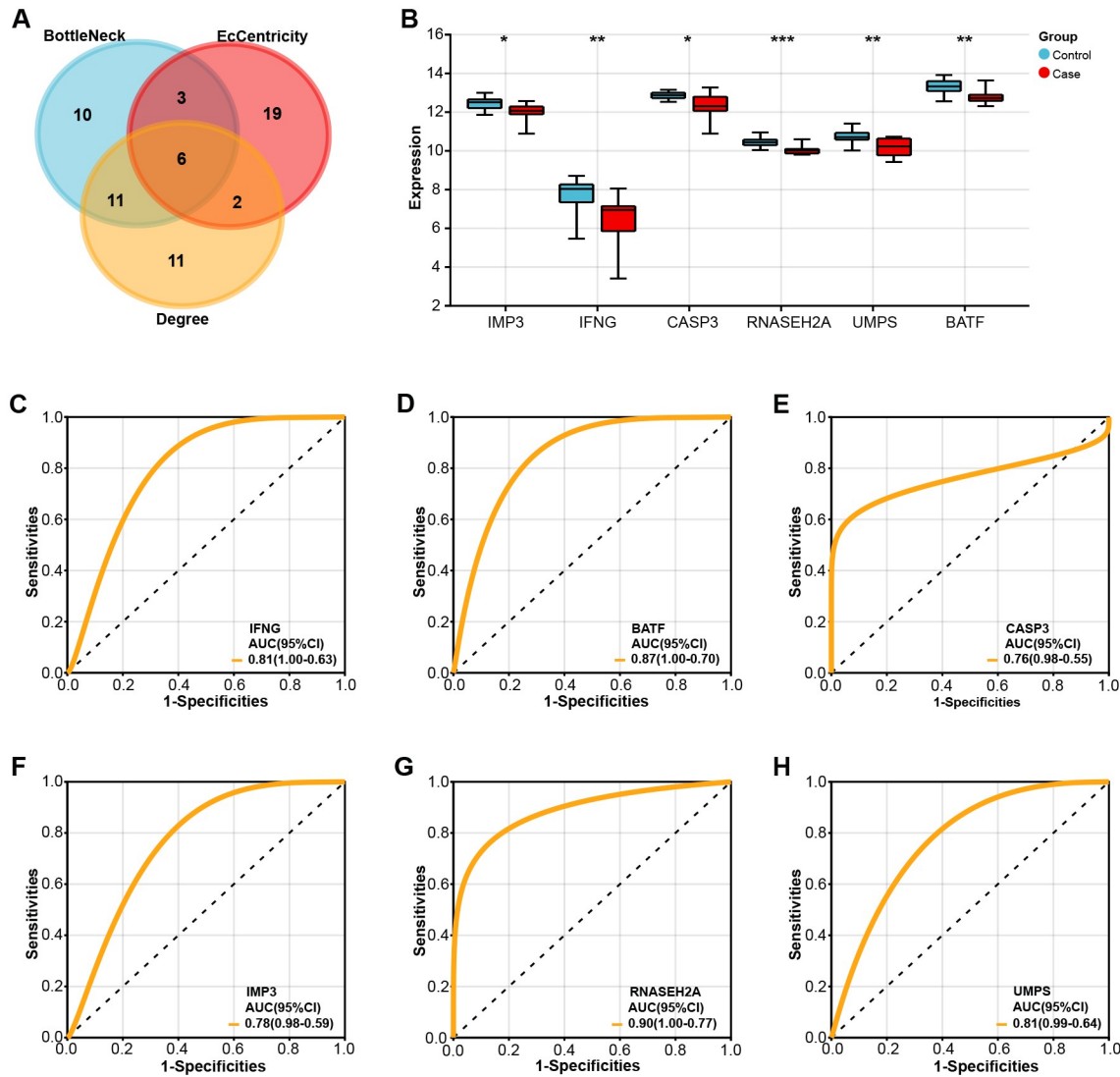

**Fig 3. Identification and validation of overlapping genes with potential clinical diagnostic value for CHD.** (A) Venn diagram of six overlapping genes, namely IMP3, IFNG, CASP3, RNASEH2A, UMPS, and BATF, identified from the PPI networks generated by the three algorithms. (B) Bar graph, significant downregulation of these six genes in case groups compared to controls in the GSE42148 dataset. *$p<0.05$, **$p<0.01$, ***$p<0.001$. (C-H) ROC curve analysis of six genes (C) IFNG, (D) BATF, (E) CASP3, (F) IMP3, (G) RNASEH2A, and (H)UMPS, the horizontal axis is 1-Specificities, and the vertical axis is Sensitivities. The AUC value under the ROC curve is used to evaluate the performance of the model, and the closer to 1, the better the model performance.

overexpression and si-BATF#1 were significant) (Fig 4C and 4D). Flow cytometry analysis of HCAECs under various treatment conditions showed that, following ox-LDL therapy, the rate of apoptosis rose considerably in comparison to the control group. On the other hand, the combination of ox-LDL with BATF knockdown led to a significant increase in the rate of apoptosis, although this increase was decreased when paired with overexpression of BATF (Fig 4E). Additionally, WB was used to assess the expression levels of two proteins linked to apoptosis in HCAECs: Bcl-2 and Bax (Fig 4F). The reduction in Bcl-2 protein levels in HCAECs caused by ox-LDL was a result of BATF downregulation, whereas the overexpression of BATF resulted in an increased expression of Bcl-2 under the same conditions. The levels of Bax

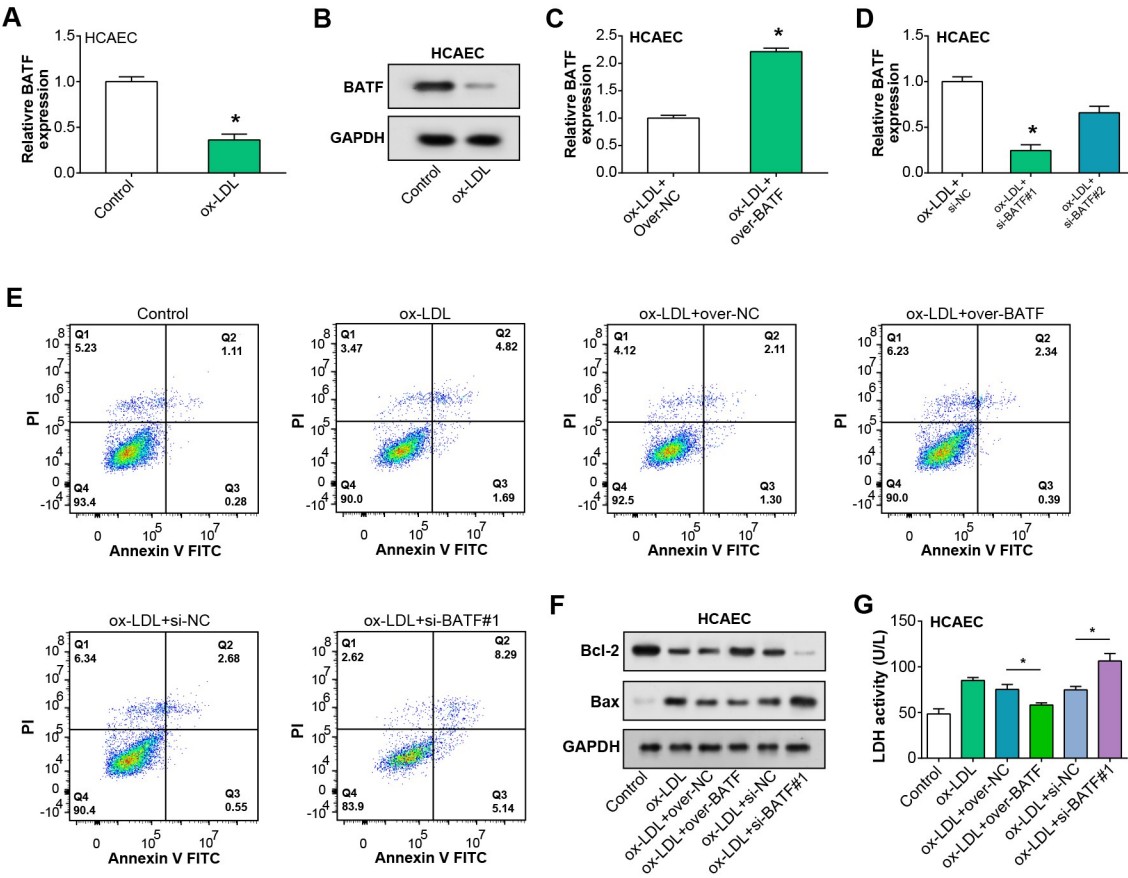

**Fig 4. BATF alleviates the damage caused by ox-LDL induction in HCAEC.** (A) qRT-PCR analysis of BATF mRNA expression in HCAEC cells following ox-LDL treatment. (B) WB analysis of BATF protein levels in HCAEC cells after ox-LDL treatment. (C) Efficiency of BATF overexpression in HCAEC cells post ox-LDL treatment, as assessed by qRT-PCR. (D) Efficiency of BATF knockdown in HCAEC cells post ox-LDL treatment, as determined by qRT-PCR. (E) Flow cytometry analysis of apoptosis in HCAEC cells post ox-LDL treatment, with either BATF overexpression or knockdown. Quadrants represent different stages of cell death: Lower left, live cells; lower right, early apoptotic cells; upper right, late apoptotic cells; upper left, necrotic cells. (F) WB analysis of apoptosis-related proteins Bcl-2 and Bax in HCAEC cells after ox-LDL treatment and BATF overexpression or knockdown. (G) LDH activity in homogenates of HCAEC cells post ox-LDL treatment and either BATF overexpression or knockdown, as measured by the LDH assay kit. *$p<0.05$.

protein displayed an inverse trend to that of Bcl-2. Finally, the LDH activity in HCAECs, as measured by the LDH assay kit, indicated that overexpression of BATF reduced the ox-LDL-induced LDH activity, while BATF knockdown amplified the ox-LDL-induced LDH activity (Fig 4G). Therefore, our findings suggested that BATF ameliorates the injury of HCAEC by ox-LDL to some extent.

## BATF regulates SIRT1 expression in ox-LDL-treated HCAEC

SIRT1 mRNA and protein levels were shown to be elevated in HCAEC following treatment with ox-LDL, respectively, according to qRT-PCR and WB analysis (Fig 5A and 5B). The DNA-binding sequence of BATF was obtained from the online JASPAR database. The correlation diagram (Fig 5C) illustrates the predicted binding sites between BATF and SIRT1. The dual-luciferase reporter assay revealed heightened luciferase activity in cells co-transfected with overexpressed BATF and SIRT1 3'UTR WT, suggesting a potential enhancement of

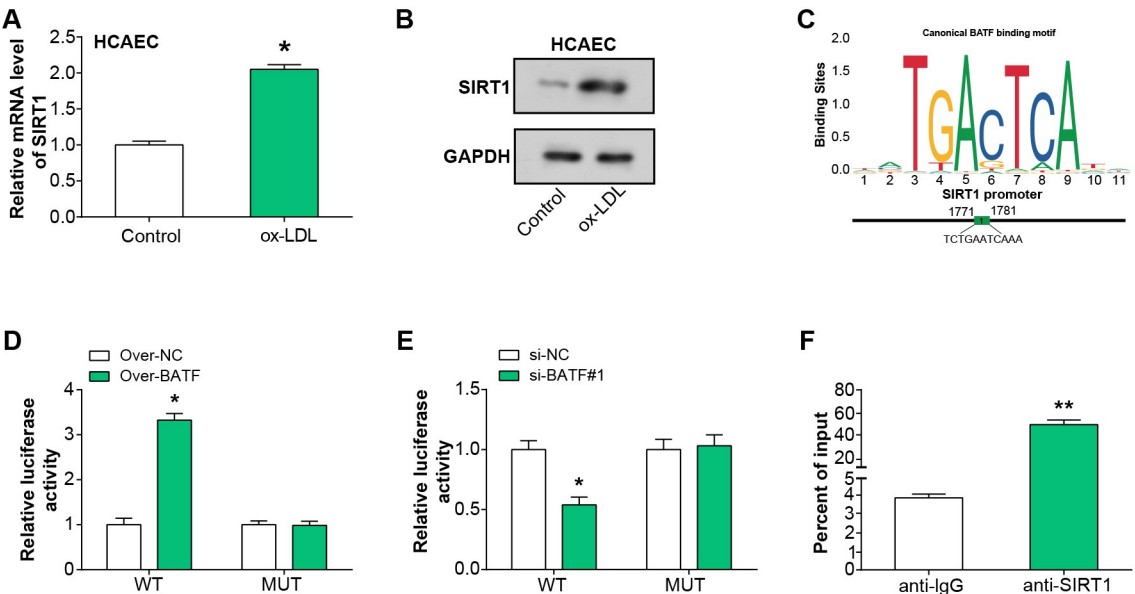

**Fig 5. BATF transcriptionally activates SIRT1 expression in ox-LDL-treated HCAEC.** (A) qRT-PCR analysis of SIRT1 mRNA expression level in HCAEC after ox-LDL treatment. (B) WB analysis detected the expression level of SIRT1 protein in HCAEC after ox-LDL treatment. (C) The correlation map of the JASPAR database shows the DNA-binding motif of BATF and the binding site with SIRT1. (D) Dual-luciferase reporter assay demonstrates luciferase activity in cells co-transfected with BATF overexpression and SIRT1 3'UTR WT. (E) Dual luciferase reporter assays luciferase activity in cells co-transfected with si-BATF#1 and SIRT1 3'UTR WT. (F) ChIP assay detected DNA affinity for BATF and SIRT1 promoter regions. *$p<0.05$, **$p<0.01$.

SIRT1 promoter activity by BATF overexpression. The increase in promoter activity was not as pronounced in the mutant promoter (Fig 5D). Conversely, si-BATF#1 and SIRT1 3'UTR WT co-transfected cells showed decreased luciferase activity, indicating that BATF knock-down could suppress SIRT1 expression (Fig 5E). In addition, ChIP assays showed that BATF had a strong affinity to the promoter region of SIRT1, further confirming that BATF may be involved in the transcriptional regulation of SIRT1 (Fig 5F). All of these findings pointed to the possibility that BATF controls SIRT1 expression in HCAEC cells treated with ox-LDL.

## BATF affects ox-LDL-induced HCAEC apoptosis by regulating SIRT1 expression

To delve deeper into the functional interaction between BATF and SIRT1 in ox-LDL-induced apoptosis, we conducted flow-through experiments on HCAEC cells. These cells were sub-jected to ox-LDL treatment and then manipulated for BATF expression—either overexpressed or knocked down—followed by the administration of SIRT1 inhibitors or activators. We also included a control group where cells were manipulated for BATF expression alongside DMSO treatment. Our findings revealed intriguing results. When cells overexpressing BATF were treated with the SIRT1 inhibitor (EX527), there was a significant increase in apoptosis com-pared to cells overexpressing BATF alone (Fig 6A). Conversely, the apoptotic rate decreased in ox-LDL-treated HCAEC cells when BATF was knocked down and treated with the SIRT1 acti-vator (S1129) (Fig 6B). Furthermore, we evaluated LDH activity in the supernatants of treated cells as a marker of cellular damage and death. Consistent with the apoptosis data, cells overex-pressing BATF and treated with SIRT1 inhibitors showed increased LDH activity, indicating enhanced cellular damage (Fig 6C). Conversely, cells with knocked down BATF and treated

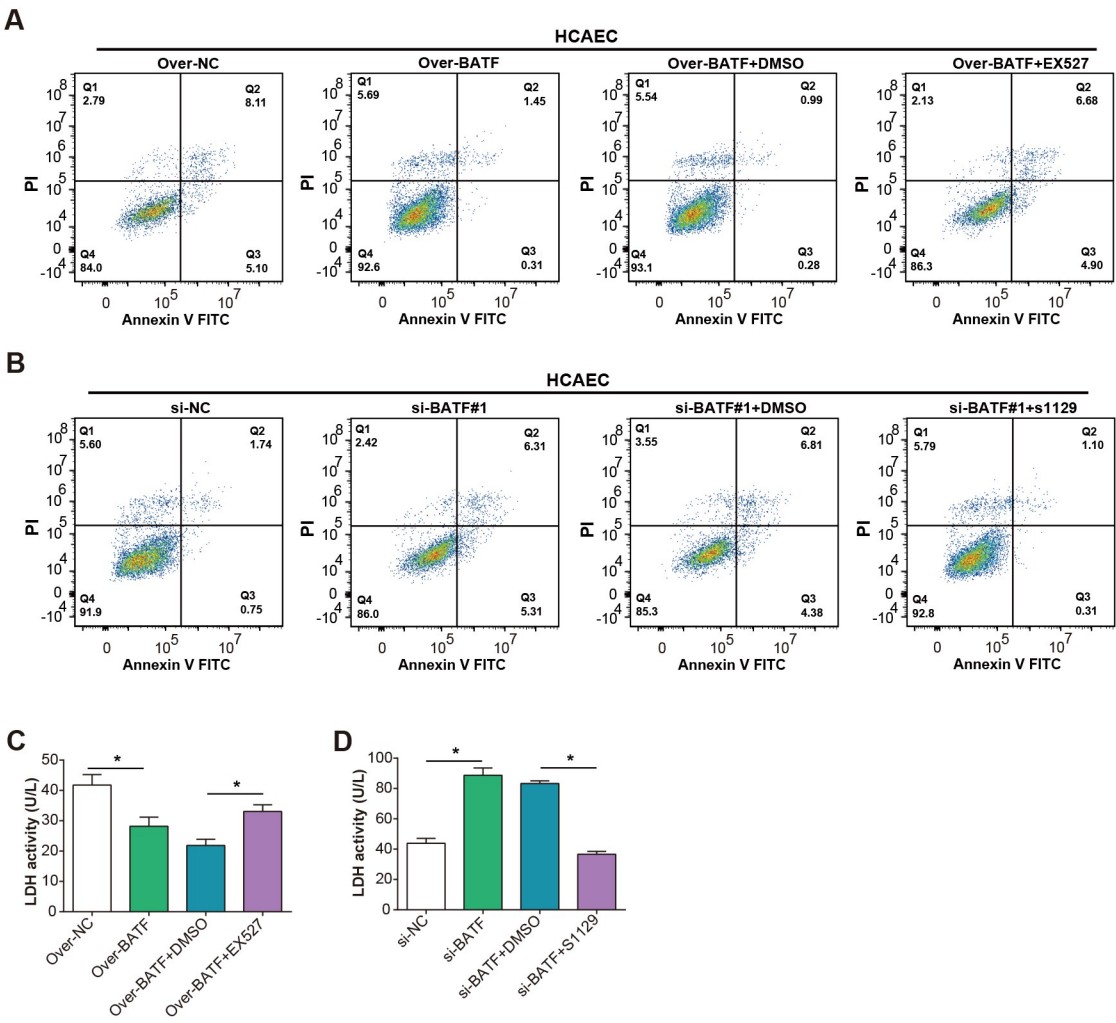

**Fig 6. Interaction between BATF and SIRT1 regulates ox-LDL-induced apoptosis and cell injury in HCAEC.** (A) The apoptosis of HCAEC with OE-BATF+SIRT1 inhibitor was detected by flow cytometry after ox-LDL treatment. (B) Flow cytometry analysis detected apoptosis of HCAECs treated with si-BATF#1+SIRT1 activator after ox-LDL treatment. (C) LDH kit detects LDH activity of homogenized cells in HCAEC with OE-BATF+SIRT1 inhibitor after ox-LDL treatment. (D) The LDH kit detects the LDH activity of homogenized cells in HCAEC with si-BATF#1+SIRT1 activator after ox-LDL treatment.

with a SIRT1 activator exhibited reduced LDH activity, suggesting decreased cellular damage (Fig 6D). Overall, these results underscored the intricate interplay between BATF and SIRT1 in regulating ox-LDL-induced apoptosis and cellular injury in HCAECs. Our inclusion of DMSO controls confirmed the specificity of these observed effects.

## Discussion

Despite advances in treatment strategies, CHD remains a significant burden on health systems worldwide [26]. Endothelial dysfunction, atherosclerosis, and myocardial infarction symptoms are the result of a complex interaction between genetic, environmental, and behavioral variables throughout its progression [27]. In addressing this challenge, activation of the NRF2/FPN1 pathway is considered an effective strategy that is expected to attenuate myocardial ischaemia-reperfusion injury in diabetic rats by modulating iron homeostasis and iron death

[28]. Meanwhile, reducing miRNA-26a-5p levels was shown to induce apoptosis in coronary endothelial cells by inhibiting the PI3K/AKT pathway [29]. In addition, the regulatory role of NR3C2 involved modulating the activation of NLRP3 inflammatory vesicles, which mediated oxidised LDL-induced endothelial cell dysfunction in human coronary arteries [30]. Meanwhile, the mechanism of action of MiR-145-5p involves mitigating hypoxia/reoxygenation-induced damage to cardiac microvascular endothelial cells in coronary artery disease by inhibiting Smad4 expression [31]. Furthermore, understanding these molecular mechanisms offers promising avenues for the development of targeted therapies aimed at mitigating the progression of coronary heart disease and improving patient outcomes.

Based on the bioinformatics findings examined, a clear link between pathology-related pathways and CHD has emerged. By performing WGCNA on DEGs in the GSE42148 gene expression profile, we classified 2044 DEGs into eight distinct modules. Notably, the black and turquoise modules showed the strongest association with the sample group and warranting further investigation. Strikingly, pathways including oligodendrocyte specification and differentiation as well as MECP2 and related Rett syndrome appeared enriched in the black module, prompting more in-depth explorations of potential neurocardiac interactions as well as the role of neuronal elements in cardiovascular pathology. Similarly, genes located in the turquoise module were greatly enriched for pathways related to apoptosis, such as nanomaterial-induced apoptosis and the regulation of apoptosis by HSP70, which may be key to unraveling the complex mechanisms of cell death and survival in coronary arteries. TOR signaling system, which is involved in cell growth, survival, and proliferation and links disruption of cellular homeostasis to atherosclerotic process and consequent cardiac injury, appears to be involved [32,33]. In the context of CHD, the molecular roles and mechanisms of these pathways may underpin our understanding of the disease and may herald innovative therapeutic strategies.

BATF and SIRT1 have become key molecules in cardiovascular research due to their unique role in cellular process activity [34]. Their distinct roles in cellular processes, particularly in cell survival, proliferation, and apoptosis, emphasize their potential relevance in the broader context of cardiovascular diseases [35]. As research continues to advance, the insights gleaned from the study of these molecules may offer promising avenues for understanding and potentially treating various cardiac conditions [36]. BATF, a transcription factor, is involved in a myriad of cellular signaling pathways. It serves as a modulator, particularly in cellular responses to oxidative stress [37]. Remarkably, recent studies have revealed its important role in regulating cellular functions. It has been reported that by genetically targeting the BATF family transcription factors BATF and BATF3 in a mouse model, we successfully observed the elimination of effector T-cell activity and achieved long-term survival of cardiac allografts [38]. On the other hand, SIRT1, a NAD-dependent deacetylase, is widely recognized for its protective role in cellular survival, proliferation, and stress responses [39]. In cardiovascular disease, the activation and upregulation of SIRT1 provide cardiac protection against a range of pathogenic mechanisms, such as oxidative stress and inflammation [40]. Inhibition of microRNA-323-3p inhibits vascular endothelial cell apoptosis by promoting sirtuin-1 expression in coronary artery disease [41]. Diosgenin prevents coronary heart disease by reducing oxidative stress and inflammation levels through Sirt1/Nrf2 and p38 MAPK pathways [42]. Building on this foundation, our recent investigations further illuminate the intricate relationship between these molecules in a specific cellular context. In our findings, overexpression of BATF accentuated ox-LDL-induced apoptosis in HCAEC, an effect that was further intensified by SIRT1 inhibition. Conversely, when BATF was knocked down and SIRT1 activated, the apoptotic effects of ox-LDL were markedly reduced, mirrored by a decrease in LDH activity, a key indicator of cellular damage. This underscores a nuanced interplay between BATF and SIRT1, particularly in the face of oxidized-LDL-induced stress. Nevertheless, our study has some

limitations, such as the elucidation of the exact mechanistic pathways and potential intermediate molecules that require further comprehensive studies to explore their translational potential in clinically relevant models, thereby harnessing the therapeutic power of modulating the BATF-SIRT1 axis for vascular lesions.

## Conclusion

In conclusion, our study revealed that BATF had the potential to act as a protective factor against ox-LDL-induced apoptosis, thereby ameliorating cellular injury. As well as the key role it played in the regulatory mechanism of SIRT1 expression in HCAECs, it provided a new perspective and promising therapeutic targets for the study of the molecular mechanisms of CHD treatment and prevention.

## Supporting information

**S1 Raw images. Original uncropped Western blot images.**
(PDF)

## Acknowledgments

The authors thank all patients involved in this study.

## Author Contributions

**Conceptualization:** Bei Tian.

**Funding acquisition:** Jingyu Ji.

**Investigation:** Bei Tian.

**Methodology:** Bei Tian, Jingyu Ji.

**Supervision:** Jingyu Ji, Can Jin.

**Validation:** Jingyu Ji, Can Jin.

**Writing – original draft:** Bei Tian.

**Writing – review & editing:** Can Jin.

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
