## [Decision Letter · Decision Letter 0]

18 Mar 2024

PONE-D-24-01735BATF alleviates ox-LDL-induced HCAEC injury by regulating SIRT1 expression in coronary heart diseasePLOS ONE

Dear Dr. Jin,

Thank you for submitting your manuscript to PLOS ONE. After careful consideration, we feel that it has merit but does not fully meet PLOS ONE’s publication criteria as it currently stands. Therefore, we invite you to submit a revised version of the manuscript that addresses the points raised during the review process.

** **==============================

We look forward to receiving your revised manuscript.

Kind regards,

Antoine Fakhry AbdelMassih

Academic Editor

PLOS ONE

Journal Requirements:

3. Thank you for stating the following financial disclosure: "Minsheng Research Project of Pudong New Area Science and Technology Development Fund(PKJ2023-Y13),Health Science and Technology Project of Pudong New Area Health Commission (PW2022A-01),Key Discipline Group of Pudong New Area Health and Health Commission (PWZxq2022-11), Peak Discipline Construction of Shanghai Pudong New Area Health Committee (PWYgf2021-04)"

Reviewers' comments:

Reviewer's Responses to Questions

**Comments to the Author**

1. Is the manuscript technically sound, and do the data support the conclusions?

Reviewer #1: Yes

Reviewer #2: Yes

2. Has the statistical analysis been performed appropriately and rigorously? 

Reviewer #1: Yes

Reviewer #2: N/A

3. Have the authors made all data underlying the findings in their manuscript fully available?

Reviewer #1: Yes

Reviewer #2: Yes

4. Is the manuscript presented in an intelligible fashion and written in standard English?

Reviewer #1: Yes

Reviewer #2: Yes

5. Review Comments to the Author

Reviewer #1: I would like to congratulate the authors on their sophisticated work, which will help unraveling some of the secrets of coronary atherosclerosis.

However, I have several comments and points of improvement:

1-Discussion:

The authors did not discuss their results thoroughly or compared their work with the existing literature. Instead, the discussion was a repetition of the introduction, emphasizing on the value of research in this area and the existing gaps of knowledge.

2- Results:

the last paragraph (just before the discussion section) is not clear, and there appears to be some mix-up in the presented information (especially in the sentence starting with "Interestingly"). I think you should revise the whole section, and re-write it in a clear and unambiguous way for readers to comprehend.

3-Conclusion:

This section is too long, and starts with an opening general statement. I believe it should be much shorter and more concise, and delver the most important finding of this research work.

Reviewer #2: The reviewer should modify the title and abstract to elucidate the abbreviations mentioned. The reviewer couldn't understand the meaning of abbreviations presented especially HCAEC.

6. PLOS authors have the option to publish the peer review history of their article (what does this mean?). If published, this will include your full peer review and any attached files.

Reviewer #1: **Yes: **Ahmed Adel Elamragy

Reviewer #2: No

---

## [Author Response · Author response to Decision Letter 0]

19 May 2024

Thank you for stating the following financial disclosure: "Minsheng Research Project of Pudong New Area Science and Technology Development Fund(PKJ2023-Y13),Health Science and Technology Project of Pudong New Area Health Commission (PW2022A-01),Key Discipline Group of Pudong New Area Health and Health Commission (PWZxq2022-11), Peak Discipline Construction of Shanghai Pudong New Area Health Committee (PWYgf2021-04)"

Response: Thank you for your review and valuable suggestions. In response to your proposed modifications regarding the funder acknowledgment, we have made the necessary adjustments and included the revised statement in our article. The modified statement is as follows: "We received key support from funders above for this study. Funding financial support from the Pudong Science and Technology Development Fund and the Pudong Health and Wellness Committee provided the necessary financial security to enable the study to proceed smoothly. These funds were used for study design, data collection and analysis, and preparation of relevant manuscripts. The support of the funders was instrumental in facilitating the progress and success of this study. We sincerely thank them for their generous support and trust." Once again, we appreciate your guidance and attention.

Review Comments to the Author

Reviewer #1: I would like to congratulate the authors on their sophisticated work, which will help unraveling some of the secrets of coronary atherosclerosis.

However, I have several comments and points of improvement:

1-Discussion:

The authors did not discuss their results thoroughly or compared their work with the existing literature. Instead, the discussion was a repetition of the introduction, emphasizing on the value of research in this area and the existing gaps of knowledge.

Response: Thank you for your valuable comments. We value your feedback on the discussion section of our paper. We do realize that in the discussion section we did not adequately discuss the results or compare our work with the existing literature. In our revision, we will ensure that our results are analyzed in more depth and compared to the existing literature to better highlight the contributions and innovations of our study. We have revisited the discussion section to ensure that it is different from the introduction section and focuses on the interpretation and analysis of our results, as well as ensuring that it is compared to the relevant literature to better reflect the originality and significance of our study. Once again, thank you for your guidance and suggestions, which we will take seriously and do our best to ensure the quality and integrity of the paper.

2- Results:

the last paragraph (just before the discussion section) is not clear, and there appears to be some mix-up in the presented information (especially in the sentence starting with "Interestingly"). I think you should revise the whole section, and re-write it in a clear and unambiguous way for readers to comprehend.

Response: Thank you for your feedback on my thesis. I will carefully consider your suggestions and revise the results section to improve clarity and accuracy. I have revisited the entire paragraph and ensured that the information is presented clearly and concisely to avoid confusing the reader. Thank you again for your review and valuable comment.

3-Conclusion:

This section is too long, and starts with an opening general statement. I believe it should be much shorter and more concise, and delver the most important finding of this research work.

Response: Thank you for your feedback and suggestions. I fully understand your concerns about this section being too long and starting off too general. I have revisited this section and made it more concise to ensure that it more directly conveys the most important findings of this study. I have eliminated the lengthy content and focused on presenting the core ideas to ensure that readers can clearly understand our findings. Once again, thank you for your valuable comment.

Reviewer #2: The reviewer should modify the title and abstract to elucidate the abbreviations mentioned. The reviewer couldn't understand the meaning of abbreviations presented especially HCAEC.

Response: Thank you very much for your valuable comments and review. I fully understand your concern about the identification of abbreviations, especially for the meaning of HCAEC. I will revise the title and abstract of the paper to explain more clearly all the abbreviations used, including HCAEC. During the revision process, I will make sure that the title and abstract accurately reflect the abbreviations used so that readers can more easily understand what our study is about. Thank you again for your suggestions and patient guidance.

---

## [Decision Letter · Decision Letter 1]

20 Jun 2024

BATF alleviates ox-LDL-induced HCAEC injury by regulating SIRT1 expression in coronary heart disease

PONE-D-24-01735R1

Dear Dr. Tian,

We’re pleased to inform you that your manuscript has been judged scientifically suitable for publication and will be formally accepted for publication once it meets all outstanding technical requirements.

Kind regards,

Antoine Fakhry AbdelMassih

Academic Editor

PLOS ONE

Additional Editor Comments (optional):

Reviewers' comments:

Reviewer's Responses to Questions

**Comments to the Author**

1. If the authors have adequately addressed your comments raised in a previous round of review and you feel that this manuscript is now acceptable for publication, you may indicate that here to bypass the “Comments to the Author” section, enter your conflict of interest statement in the “Confidential to Editor” section, and submit your "Accept" recommendation.

Reviewer #1: (No Response)

2. Is the manuscript technically sound, and do the data support the conclusions?

Reviewer #1: Yes

3. Has the statistical analysis been performed appropriately and rigorously? 

Reviewer #1: Yes

4. Have the authors made all data underlying the findings in their manuscript fully available?

Reviewer #1: Yes

5. Is the manuscript presented in an intelligible fashion and written in standard English?

Reviewer #1: Yes

6. Review Comments to the Author

Reviewer #1: Thank you for responding to our observations. However, the discussion section still needs some improvement. It should begin with the most important finding of your experiment. The discussion should also be free of the general information (that is more appropriate for the introduction). Instead, please focus on the findings of your research, and their strengths and limitations, comparing them with other research in the same field.

7. PLOS authors have the option to publish the peer review history of their article (what does this mean?). If published, this will include your full peer review and any attached files.

Reviewer #1: **Yes: **Ahmed Adel Elamragy

---

## [Editor Report · Acceptance letter]

5 Dec 2024

PONE-D-24-01735R1 

PLOS ONE

Dear Dr. Jin, 

I'm pleased to inform you that your manuscript has been deemed suitable for publication in PLOS ONE. Congratulations! Your manuscript is now being handed over to our production team.

Kind regards, 

on behalf of

Prof Antoine Fakhry AbdelMassih 

Academic Editor

PLOS ONE